# A Micron-Range Displacement Sensor Based on Thermo-Optically Tuned Whispering Gallery Modes in a Microcapillary Resonator

**DOI:** 10.3390/s22218312

**Published:** 2022-10-29

**Authors:** Zhe Wang, Arun Kumar Mallik, Fangfang Wei, Zhuochen Wang, Anuradha Rout, Qiang Wu, Yuliya Semenova

**Affiliations:** 1Photonics Research Centre, School of Electrical and Electronic Engineering, Technological University Dublin, D07 ADY7 Dublin, Ireland; 2Tyndall National Institute, University College Cork, Lee Maltings, Dyke Parade, T12 YN60 Cork, Ireland; 3Department of Mathematics, Physics and Electrical Engineering, Northumbria University, Newcastle upon Tyne NE1 8ST, UK

**Keywords:** whispering gallery modes, capillary, displacement sensor, thermo-optic tuning

## Abstract

A novel micron-range displacement sensor based on a whispering-gallery mode (WGM) microcapillary resonator filled with a nematic liquid crystal (LC) and a magnetic nanoparticle- coated fiber half-taper is proposed and experimentally demonstrated. In the proposed device, the tip of a fiber half-taper coated with a thin layer of magnetic nanoparticles (MNPs) moves inside the LC-filled microcapillary resonator along its axis. The input end of the fiber half-taper is connected to a pump laser source and due to the thermo-optic effect within the MNPs, the fiber tip acts as point heat source increasing the temperature of the LC material in its vicinity. An increase in the LC temperature leads to a decrease in its effective refractive index, which in turn causes spectral shift of the WGM resonances monitored in the transmission spectrum of the coupling fiber. The spectral shift of the WGMs is proportional to the displacement of the MNP-coated tip with respect to the microcapillary’s light coupling point. The sensor’s operation is simulated considering heat transfer in the microcapillary filled with a LC material having a negative thermo-optic coefficient. The simulations are in a good agreement with the WGMs spectral shift observed experimentally. A sensitivity to displacement of 15.44 pm/µm and a response time of 260 ms were demonstrated for the proposed sensor. The device also shows good reversibility and repeatability of response. The proposed micro-displacement sensor has potential applications in micro-manufacturing, precision measurement and medical instruments.

## 1. Introduction

A displacement sensor is a device used for the measurement of the positional movement of a given object. Depending on the operating principle, there are different types of displacement sensors including resistive, capacitive, inductive, ultrasonic, piezo, electromagnetic, and optical [1,2,3,4]. Displacement sensors are widely used in various fields such as motorsport, automotive, industrial process control applications, structural health monitoring, agriculture, aerospace, robotics and many others [5,6,7]. Most of traditional electronic displacement sensors rely on measurements of resistance, capacitance or eddy currents converting these quantities into information about the object’s position [4]. Typically, electronic displacement sensors suffer from poor linearity and high susceptibility to environmental and electromagnetic interferences. Optical fiber displacement sensors are very attractive due to their light weight, high sensitivity, fast response, and immunity to electromagnetic interference. A variety of fiber optic displacement sensors have been proposed and developed based on microfiber couplers [8], Mach-Zehnder [9], or Fabry-Perot interferometers [10]. Utilizing interference between light from different paths, the above-mentioned structures can accurately sense the position of the object by converting the light path difference into a spectral shift or an intensity change. To enhance their sensitivity, interferometric sensing configurations frequently utilize Vernier effect, which is realized by cascading two interferometers [11].

Whispering gallery mode (WGM) resonators are micron-sized dielectric structures with a circular symmetry. Typically, light is coupled into such a resonator from an external evanescent light field (of a tapered fiber) and is trapped within the microcavity due to almost total internal reflections at the boundary of the resonator and its surrounding. Within the microcavity, constructive interference takes place for the trapped light of specific optical wavelengths that meet the resonance condition *mλ* = 2*πRn_eff_*, where *m* is an integer, *λ* is the resonant wavelength, *R* is the radius of the microcavity and *n_eff_* is its effective refractive index. Light at wavelengths that do not meet the resonance condition is not trapped in the cavity and is transmitted to the coupling taper. Given the high-quality factors and very high sensitivity of the WGM resonances to the resonator size/refractive index, such resonators offer an alternative way to realize displacement measurements with ultrahigh sensitivity and excellent measurement resolution.

Several previously reported WGM-based sensors proved to be capable of measurement micron-scale displacements. One such displacement sensor reported in [12] involves compacting two polystyrene microsphere resonators inside a capillary within a section of a micro-structured optical fiber. Variation of the spacing between the two microspheres induced by stretching the capillary as a result of pulling of its free end, leads to complex changes in the WGM spectrum of the microresonators which is correlated with the displacement value. Such a double-cavity resonator sensor could potentially offer a nanometer order measurement resolution in a narrow range of displacements from 0 to 15 nm, but excitation of the WGMs simultaneously in two microresonators greatly increases system complexity and its reproducibility is difficult to achieve in a practical device. An alternative approach involves conversion of the applied displacement into bending of the fiber used to couple light inside the WGM resonator, so that the resulting change in the polarization state of the coupled light and corresponding changes in the WGM spectrum are linked to the value of the displacement [13]. This type of sensor has a wide measurement range of over 400 μm, but a limited sensitivity (33 dB/mm) and measurement resolution of just ~10 μm. Another WGM-based displacement sensor was proposed in [14] where a silica microbubble resonator packaged within a Polydimethylsiloxane (PDMS) casing was used together with a solid microsphere, pressed into the PDMS packaging using a nano-positioning stage. Although the sensor had excellent long-term stability its displacement measurement range is limited by the thickness of the PDMS packaging (20 μm) with maximum sensitivity of 0.1 pm/µm. The above reports demonstrated high potential of the WGM-based displacement sensors and also highlight the need for further improvement of their performance particularly in terms of sensitivity and resolution in a wider range of displacements combined with good repeatability and long-term stability.

Recently, we proposed and experimentally demonstrated thermo-optic tuning of a microcapillary resonator filled with a nematic liquid crystal (LC) and a fiber half-taper coated with a thin layer of magnetic nanoparticles (MNPs) placed inside the resonator [15]. In this work, we show that the above structure can be utilized as a displacement sensor with high sensitivity of 15.44 pm/µm in the displacement range of 200 µm. The proposed sensor offers good repeatability, small hysteresis and a competitive response time of 260 ms. In this work, we also simulate the electric field distribution within the fiber half-taper and temporal evolution of the heat transfer between the MNPs-coated fiber half-taper and the LC in the capillary using the finite element method. The simulated sensor response is in a good agreement with the experimental results. Compared with other WGM resonator-based displacement sensors utilizing pulling or bending, our proposed sensor offers better mechanical stability and ease of manipulation and adjustment. To the best of our knowledge, this is the first attempt to utilize the photothermal effect within a WGM microcapillary to realize a displacement sensor. Our proposed device shows excellent potential for applications in micro-manufacturing, precision measurement and medical instruments where measurements of displacement on a micron scale are required.

In a broader sense, recent developments in nonlinear and quantum optics indicate growing interest in WGM resonators for a range of applications including WGM microlasers [16], rapidly developing field of optomechanics, that focuses of interactions of WGM photons with acoustic phonons [17], and utilizing WGM resonators for achieving strong interaction between individual photons [18], which would constitute a breakthrough in quantum logic, quantum computing and precision measurement with photons.

## 2. Materials and Methods

A tapered fiber and a thin-wall capillary for our experiments were fabricated from a standard single-mode fiber (SMF-28 from Corning Inc., USA) and a commercial silica capillary (Polymicro Technologies, USA), respectively by using the customized microheater brushing technique described in [19]. In each case, the center of the single-mode fiber (or capillary) was placed in the slit of the ceramic microheater (CMH-7019, NTT-AT) where the temperature was approximately 1300 °C. A pair of translation stages driven by a customized computer program ensured the fiber/capillary were tapered down to a required diameter. In this experiment, a capillary with outer/inner dimeters of 40/36 µm and a tapered fiber with a waist diameter of 10 µm were fabricated. The fabricated microcapillary was then fixed on a glass slide using two drops of a UV curable epoxy. To fabricate a MNPs coated half-taper, the fabricated fiber taper was cut in two and one of the half-tapers was dipped into and pulled out from a solution of MNPs (Micromod Partikeltechnologie GmbH, Germany, nanomag@-D, 09-20-132, 10 mg/mL) at a rate of 0.5 mm/s. The coated half-taper was then cured at room temperature for 48 h.

For coupling of light into the microcapillary resonator another tapered fiber with a diameter of 1 µm was fabricated using the same microheater technique. Light coupling was realized by placing the fiber taper perpendicularly and in contact with the microcapillary. The ends of the fiber taper were connected to a broadband light source (BBS, Thorlabs, USA, S5FC1005S(P), 1500–1600 nm, diode current = 600 mA, FWHM = 50 nm) through a polarization controller and an optical spectrum analyzer (OSA, Advantest, Japan, Q8384), respectively. Input light polarization was varied with the help of the polarization controller while the transmission spectrum of the taper was observed on the screen of the OSA. Once a high quality WGM spectrum was achieved, the tapered fiber was fixed on a glass slide at both ends using epoxy glue.

At the next step the microcapillary was filled with the liquid crystal material, MDA-05-2782 (n_e_ = 1.6152, n_o_ = 1.4912, measured at 589.3 nm and 20 °C, clearing point: 106 °C) (Licristal, Merck, Germany), using a syringe. After the microcapillary was disconnected from the syringe, the MNPs-coated fiber half-taper was inserted in the capillary from one of its ends. The free end of the half-taper was fixed on a translation stage to realize its motion along the micropapillary axis with a micrometer resolution. The fabrication process of the displacement sensor based on thermos-optics tuning and capillary WGM resonator is illustrated in Figure 1.

The schematic diagram of the experimental setup is shown in Figure 2. The free end of the half-taper was connected to a 980 nm pump laser through one of the splitter ports (86.7%). Another port of the splitter (13.3%) was connected to an optical power meter (dBm Optics, Inc., USA, Model 4100) for monitoring of the pump laser power.

## 3. Experimental Results and Discussion

In the experiments, the position of the MNPs-coated half-taper tip within the microcapillary resonator infiltrated with LC was controlled by a micro-translation stage to realize 15 different point locations along the capillary axis (within the translation stage scale from 399 µm to −399 µm). A schematic diagram of the microcapillary with the MNPs-coated half-taper inside is shown in Figure 3a. The zero µm position of the translation stage scale corresponds to the case when the tip of the fiber half-taper is located directly under the light coupling site (line A in Figure 3a). A series of microscopic images of the device corresponding to different half-taper tip positions are shown in Figure 3b-d. The pump laser power was set to 15.6 mW and remained constant during the set of experiments.

Figure 4a shows the experimentally measured transmission spectra of the WGM microresonator corresponding to the different positions of the fiber tip within the capillary. As can be seen from the graph, the WGM spectrum blue-shifts as the tip of the half taper moves from 399 µm to circa −114 µm, while further displacement of the half-taper tip beyond the light coupling site (from −114 to −399 µm) the WGMs spectral shift becomes non-monotonic. It should be noted that there are slight variations in the shape of the WGM spectrum when the tip of the half taper moves across the light coupling site, which is likely due to some coupling of the pump laser light from the half-taper into the coupling taper.

Figure 4c illustrates the dependency of the selected spectral dip wavelength (near 1551.5 nm) versus the coordinate of the half-taper tip (black line/triangles). As can be seen from the graph, the resonant dip experiences a moderate blue shift in the range between 399 µm and 200 µm, which is followed by a “faster” blue shift within the range from 200 to 0 µm, which corresponds to the case when the tip of the fiber half-taper is on the right side of the light coupling site. This can be explained by the increase in temperature of the LC material near the light coupling site as the half-taper tip approaches the coupling point. An increase in the LC temperature results in the decrease in its effective refractive index, causing the blue shift of the WGMs. The WGM spectrum continues to shift towards shorter wavelengths but at a slower pace after the half-taper tip moves beyond the coupling site (in the range from 0 to −114 µm). This “slower” blue shift in the range from 0 to −114 µm thus can be explained by the influence of the MNPs located on the surface of the half taper further from the end tip. As expected, the strongest photothermal effect is observed when the half-taper tip is approaching the coupling site due to the proximity of the MNPs cluster acting as a heat source. The blue shift of the WGM spectrum continues when the end of the half taper moves beyond −114 µm to the left from the coupling point due to the rise in the LC temperature. Beyond circa −114 µm position of the tip, the WGM spectra experience slight red shift, likely due to a drop in temperature of the liquid crystal in the vicinity of the coupling point as the heat source moves away.

Another experiment was carried out by moving the half-taper tip in the opposite direction (from left to right) to investigate the reversibility of the sensor response. Figure 4b shows the experimentally measured transmission spectra of the same sample taken at different positions of the fiber tip moving from left to right. The dependence of the selected spectral dip wavelength (near 1551.5 nm) versus the coordinate of the half-taper tip is shown in Figure 4c (red line/triangles). As one can see from the graph, the spectral positions of the selected resonant dip versus the taper tip coordinate are almost identical when moving in the opposite directions, and although the transmission dip does not return exactly to its initial position after the full cycle of measurements, the relatively small hysteresis still demonstrates a good reversibility of the sensor. In conclusion, the WGM spectrum of the microcapillary resonator shows a very good displacement sensitivity when the half-taper tip moves in the range from 0 to 399 µm. The linear fitting of the selected spectral dips versus tip coordinate in the range from 0 to 399 µm are shown in Figure 4d. The sensitivity of the resonance dip to the taper tip coordinate in the range from 0 to 171 µm achieved in the experiment is 14 pm/µm, and the sensitivity of the taper tip coordinate in the range from 171 to 399 µm is 3.7 pm/µm.

The influence of the pump laser power level on the sensitivity of the proposed displacement sensor was also studied. These experiments were carried out at two additional pump laser powers set to 7.8 mW and 26 mW. Figure 5a shows the selected wavelength dips (at circa 1552 nm for the 7.8 mW case, and 1551 nm for the 26 mW case) versus the coordinate of the half-taper tip. As can be seen from the graph, the dependencies at different power levels follow similar trends and are consistent with the results in Figure 4c. The displacement sensitivities for the pump laser powers of 7.8 mW and 26 mW in the coordinate range from 0 µm to 171 µm were estimated as being 4.75 pm/µm and 15.44 pm/µm, and in the range from 171 µm to 399 µm as being 2.42 pm/µm and 8.41 pm/µm, respectively. This demonstrates that the sensor can be operated at lower pump laser powers but at the expense of lower sensitivity.

A preliminary study of repeatability of the sensor’s performance in time was carried out by repeating the experiment with 15.6 mW pump laser power after 5 days and indicated less than 5% difference in the displacement sensitivity.

The sensor response time was estimated using the same setup, but replacing the OSA with a photo-detector (Thorlabs, PDA10CS-EC) connected to an oscilloscope (Keysight, MSO-X 2022A). Figure 5b illustrates the voltage from the photo-detector versus time at different powers of the pump laser. Abrupt voltage changes occurred when the pump laser was turned on and off for all the cases. The average response times estimated as (reaction time + relaxation time)/2 for pump laser powers of 7.8 mW, 15.6 mW, and 26 mW are approximately 225 ms, 170 ms, and 260 ms, respectively. These results demonstrate that the proposed sensor shows good repeatability of performance and competitive response times.

To better understand the evolution of the spectrum in time, spectral shift of the selected resonance was monitored over the time interval of 70 min during which the pump laser was switched on before switching it off. Figure 6a shows the wavelength of the selected WGM resonance plotted against time for pump laser powers of 7.8 mW, 15.6 mW, and 26 mW. It should be noted that the tip of the half-taper was fixed at the position corresponding to 342 µm for all pump powers for consistency. The WGM spectra were collected with a fixed time interval of 1 min during the experiment.

As can be seen from the graph, the spectral shifts in response to switching the pump laser on are different for the different power levels and show good stability over time. It is found that the resonance wavelength shifted almost immediately after the status of the laser changed. Figure 6b illustrates the dependency of the selected spectral dip wavelength (at circa 1552.7 nm) versus the pump laser power. As can be seen from the graph, the resonance wavelength experiences a blue shift with the increase in the pump laser power, the linear fitting equation for the power increase is *y* = −0.076*x* + 1552.66.

## 4. Analysis and Simulation of the Thermo-Optic Effect in the Microcapillary

The operating principle of the proposed sensor relies on the thermo-optic effect within the layer of MNPs coated on the surface of a fiber half-taper, where the light from the pump laser is absorbed and then converted into heat. Therefore, the tip of the half-taper acts as a point heat source increasing the temperature of the LC material in its vicinity. An increase in the LC temperature leads to a decrease in its effective refractive index, which in turn causes spectral shift of the WGM resonances monitored in the transmission spectrum of the coupling fiber. Due to the nature of the heat transfer within the microcapillary, the spectral shift of the WGMs is proportional to the displacement of the MNPs-coated tip with respect to the microcapillary’s light coupling point.

To better understand the phenomena within the proposed structure and to optimize the parameters of the proposed senor, it is useful to simulate the thermo-optic effect in the microcapillary. This was carried out using COMSOL Multiphysics (5.5). At first, we simulated the distribution of the electric field within the half-taper tip illuminated with a radiation of a 980-nm laser. As one can see from Figure 7, the pump laser light is fully confined to the singlemode fiber core at the thick end of the half-taper, but the electric field intensity distribution near the fiber surface increases significantly towards the half-taper tip, especially when the diameter of the fiber half-taper decreases beyond 5 µm. Provided the surface of the taper is coated with a high density MNPs coating around the end of the half-taper, this should facilitate a strong photothermal effect.

As the next step, the heat transfer in the microcapillary was simulated using a three-dimensional model in COMSOL. In the simulation, an MNPs-coated fiber half-taper with a diameter of 12 μm was considered to be at the center of a liquid crystal-filled capillary with the outer/inner diameters of 50 μm and 46 μm, as shown in Figure 3a. Both the tapered fiber and the capillary are made from silica. For simplicity the MNPs-coating at the fiber half-taper tip was considered as a continuous boundary heat source at the interface between the fiber taper and LC. The initial temperature of the LC in the simulation was set to 17 °C. The heat transfer equation as the basis for the simulation is [20]:(1)$$\rho C_{p}\frac{\partial T}{\partial t}+\rho C_{p}u\cdot \nabla T-\nabla \left(k\cdot \nabla T\right)=Q$$
where ρ is the mass density of the LC, $C_{p}$ is the LC heat capacity, *T* is the real-time temperature of the LC, u is the fluid velocity vector, *k* is thermal conductivity of the LC and *Q* represents heat resource per volume. In this process, 3400 J/(kg·K) and 0.17 W/(m·K) for liquid-crystal octylcyanobiphenyl (8CB) are adopted as $C_{p}$ and *k* of the MDA-05-2782, respectively, due to the similarity of their properties [21]. The density of the LC was assumed as 985 kg/m^3^. The final effective boundary heat source *Q* was defined as (P × heat convection ratio)/S, where P is the laser power and S is the superficial area of MNPs-coating. We assume the length of the coating area at the end of the fiber half-taper as 100 μm. To achieve a good match with our experimental data, in the simulation we assumed that only 5% of the laser pump power was converted into heat. For simplicity we only consider the external natural convection in air cooling process, in accordance with the convective heat equation [20]:(2)$$n\cdot \left(k\cdot \nabla T\right)=h\left(T_{ext}-T\right)$$
where *n*, *k*, $T_{ext}$ and *T* are the normal vector, thermal conductivity of the LC, external temperature, and real-time temperature of the LC, respectively.

The time-dependent heat response inside the capillary was simulated assuming a continuous boundary heat source at the tip of the half-taper at 2000 ms for 4000 ms before being switched off. The temperature distribution in the capillary with the half-taper tip at the coordinate of 250 μm and at the time corresponding to 5900 ms is shown in Figure 8a.

As expected, the maximum temperature change occurred around the tip of the tapered fiber. With a continuous heat flux, the heat is efficiently transferred from the heat source to the LC due to the relatively large thermal conductivity of the LC. Along the direction of the capillary (negative direction of z-axis in Figure 8a), the temperature shows a decreasing trend as the distance increases away from the fiber half-taper tip. Figure 8b illustrates maximum temperatures within x-y cross sections at different distances away from the tip. Figure 8c illustrates a series of temperature distributions at different times after the heat source was switched on at 2000 ms. It can be seen that the boundary accumulates heat swiftly as the source is switched on so that the temperature of the LC increases fast due to its large thermal conductivity. The temperature distribution in the capillary stabilizes after continuous heat flux for 500 ms, indicating that the time response of the heat transfer is less than 500 ms, which is consistent with the experimental data in Figure 5b.

To confirm the results of our experiment in Figure 3a, another simulation was carried out where the tip of the fiber half-taper was placed at 7 different locations between 0 μm and 350 μm coordinates with an interval of 50 μm. The temperature at the coupling site (point A) in the vicinity of the capillary wall in Figure 3a is of particular interest since it determines the spectral positions of the WGM resonances in the transmission spectrum of the light coupling taper. As can be seen from Figure 9a, abrupt temperature changes at this point occurred when the heat source was switched on and off at 2000 ms and 6000 ms from the start of the simulation. Figure 9b illustrates the dependency of the temperature at point A versus the coordinate of the half-taper tip. As can be seen from the graph, the closer is the half-taper tip to the coupling site the higher is the temperature. The temperature changes are sharper when the half-taper tip is close to the coupling site and become less significant as the half-taper tip moves further away from it. As we demonstrated previously in [15], the temperature of the LC in the vicinity of the light coupling site determines the value of the effective RI of the microresonator and thus the spectral positions of the WGM resonances. Therefore, the results in Figure 9b are in agreement with the experimental data in Figure 4c given the negative thermo-optic coefficient of the LC [22]. The dependence of temperature at point A as a function of pump laser power when the half-taper is at 350 μm is shown in Figure 9c.

As expected, the temperature changes linearly with the pump power increase, leading to a decrease in the effective RI of the LC and resulting in a blue shift of the WGM resonances, which is consistent with the result in Figure 6b.

## 5. Conclusions

In conclusion, a new micro displacement sensor based on thermo-optic tuning of WGMs in a microcapillary resonator has been proposed and experimentally demonstrated. In the proposed device, the tip of a fiber half-taper coated with a thin layer of MNPs moves along the axis of the LC-filled microcapillary resonator. The input end of the fiber half-taper is connected to a pump laser source with a fixed power level and due to the thermo-optic effect within the MNPs, the fiber tip acts as a point heat source, increasing the temperature of the LC material in its vicinity. An increase in the LC temperature leads to a decrease in its effective refractive index, which in turn causes spectral shift of the WGM resonances monitored in the transmission spectrum of the coupling fiber. The spectral shift of the WGMs is proportional to the displacement of the MNPs-coated tip with respect to the microcapillary’s light coupling point. Sensitivity of the proposed device to displacement increases with the increase in the laser pump. Maximum sensitivity to displacement of 15.44 pm/µm at pump laser power of 26 mW in the range of displacements from 0 to 171 µm and response times in the order of 260 ms have been demonstrated experimentally. The device also shows good reversibility and repeatability of response. In addition, the operation of the proposed sensor was simulated using the finite element method considering the heat transfer in the microcapillary filled with a LC material with a negative thermo-optic coefficient. The simulations are in a good agreement with the WGMs spectral shift observed experimentally.

The proposed micro displacement sensor has many potential applications in micro-manufacturing, precision measurement, robotics and automation, including distance, object positioning, quality control, thickness, diameter or thickness measurements. It should be noted that a number of displacement sensor technologies exist currently and variable requirements, e.g., in relation to the object surface quality or the necessary precision, result from the specific application task. Table 1 analyses the various performance factors for the most popular technologies and provides the comparison with our proposed sensor.

As can be seen from the table, the proposed sensor has a unique combination of advantages, offering one of the highest measurement resolutions, together with small probe size, ability to measure features not visible by non-contact sensors, operation in dirty or dusty environments, immunity to electromagnetic interference, independence on the type of target material and surface type. Potential application niches are likely to be in precision structural health monitoring of engineering structures in real life conditions and process control/automation in harsh/dusty environments.

## Figures and Tables

**Figure 1 sensors-22-08312-f001:**
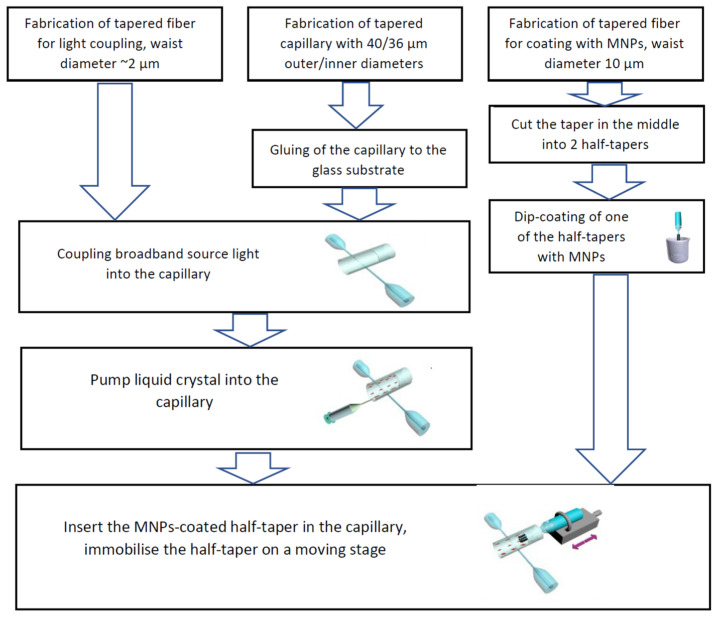
Flow chart of the fabrication process of displacement sensor based on the LC-infiltrated capillary WGM resonator.

**Figure 2 sensors-22-08312-f002:**
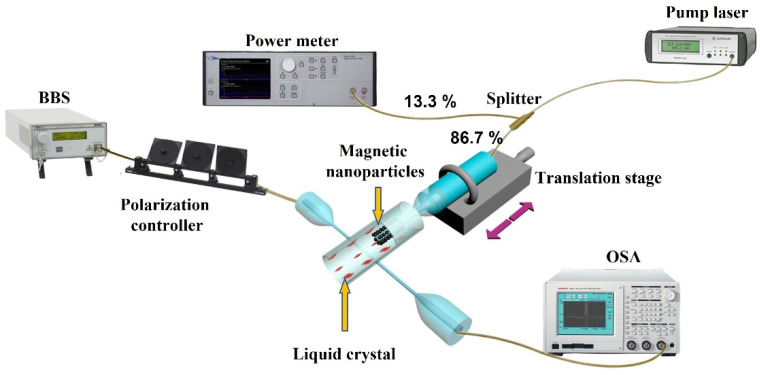
Schematic diagram of the LC-infiltrated capillary WGM resonator and experimental setup for its characterization.

**Figure 3 sensors-22-08312-f003:**
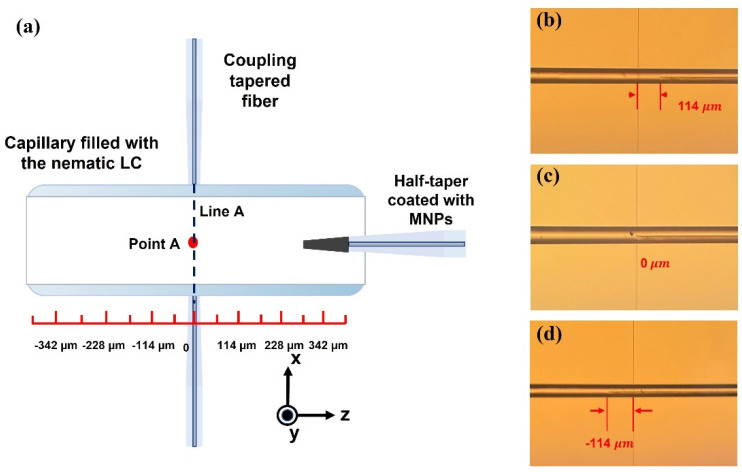
(**a**) Schematic diagram of the LC-infiltrated capillary WGM resonator and experimental setup for its characterization; (**b**–**d**) microscopic images of the capillary corresponding to different positions of the half-taper tip.

**Figure 4 sensors-22-08312-f004:**
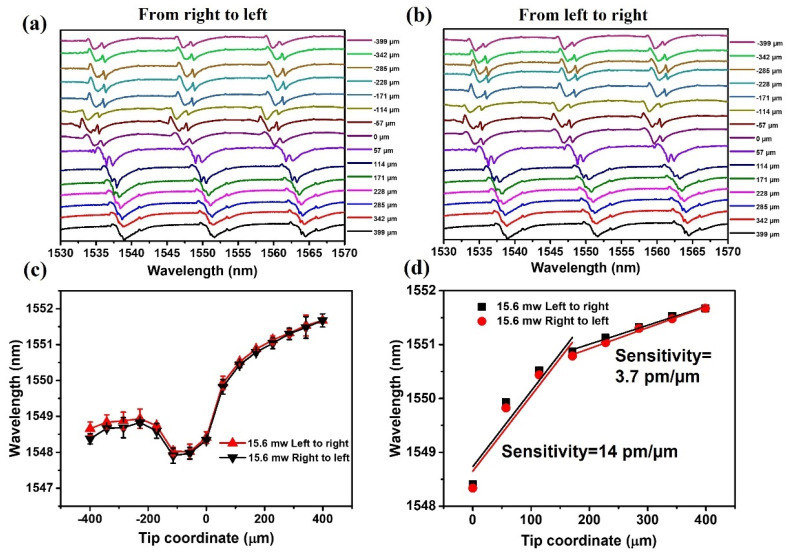
Experimental transmission spectra of the microcapillary resonator at different locations of the half-taper tip: (**a**) fiber half-taper moves from right to left, (**b**) fiber half-taper moves from left to right, (**c**) selected transmission dip wavelength versus tip coordinate, (**d**) linear fitting of the selected spectral dip wavelength versus tip coordinate.

**Figure 5 sensors-22-08312-f005:**
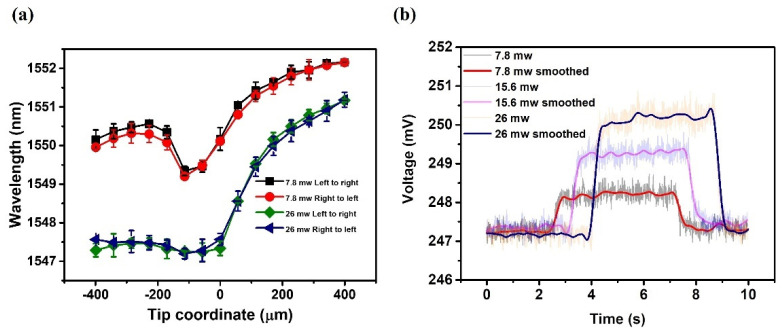
(**a**) Experimental data of the selected transmission dip wavelength versus the tip coordinate for the sample at two different pump laser powers: 7.8 mW and 26 mW, (**b**) temporal response of the sensor at different pump laser powers of 7.8 mW, 15.6 mW, and 26 mW.

**Figure 6 sensors-22-08312-f006:**
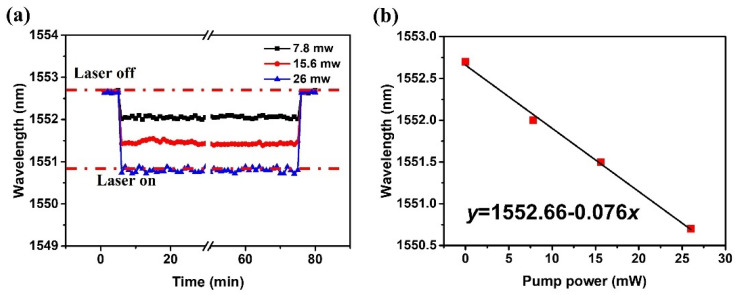
(**a**) Temporal response of the sensor at different pump laser powers of 7.8 mW, 15.6 mW, and 26 mW; (**b**) dependence of the selected transmission dip wavelength versus the pump laser power.

**Figure 7 sensors-22-08312-f007:**
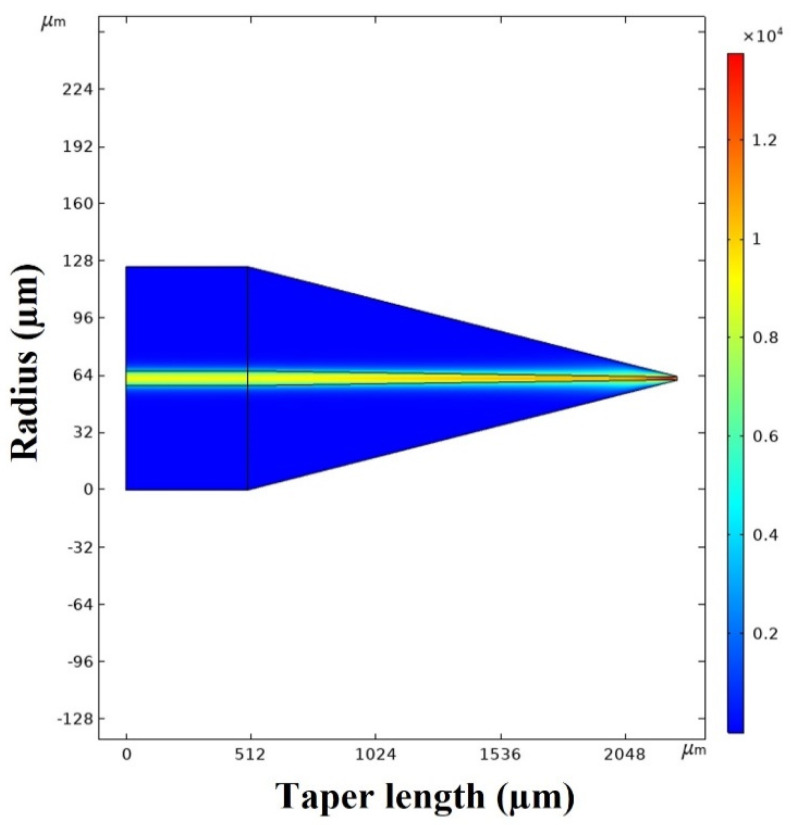
Simulated electric field intensity distribution within the half-taper end.

**Figure 8 sensors-22-08312-f008:**
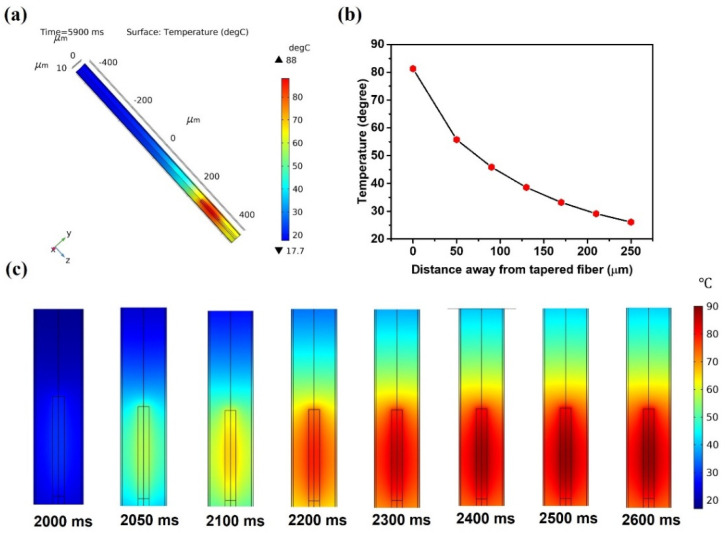
(**a**) 3D temperature distribution in the capillary at 15.6 mW laser power, (**b**) maximum temperature in x-y plane versus distance away from the half-taper tip, (**c**) 2D temperature distributions around the half-taper tip at different times for the laser power is 15.6 mW.

**Figure 9 sensors-22-08312-f009:**
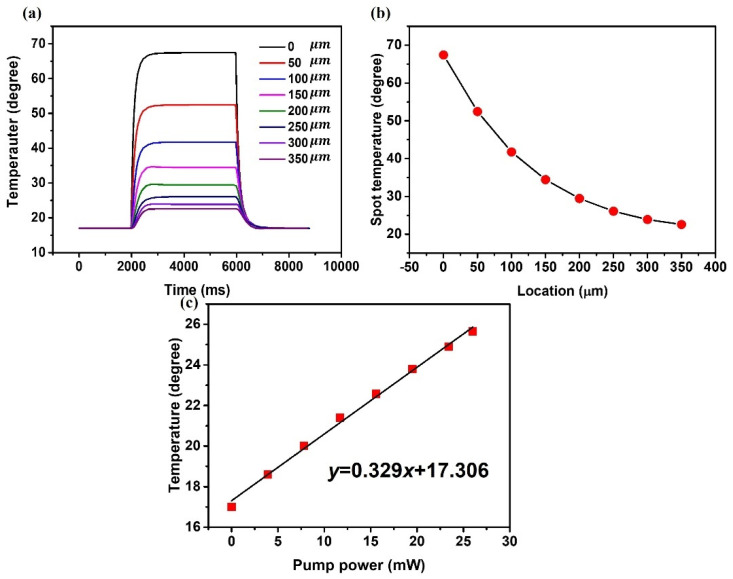
(**a**) Temperature at point A with a fiber half-taper placed at various positions between 0 and 350 µm versus time at pump power of 15.6 mW, (**b**) dependence of the maximum temperature at the coupling site (point A) versus the tip location at 15.6 mW, (**c**) temperature at the light coupling site A for the tip at 350 µm versus laser power.

**Table 1 sensors-22-08312-t001:** Comparison of different high-accuracy displacement sensor technologies.

Factor	High Accuracy Displacement Sensor Technologies					
	Non-Contact Sensors			Contact Sensors		
	Capacitive	Eddy Current	Laser/Optical	LVDTs ^1^	Fiber-Optic	This Work
High Resolution	Excellent (nm)	Ok (>µm)	Excellent (nm)	Ok (>µm)	Poor (mm)	Excellent (nm)
Small probe size (<mm)	No	No	No	No	Yes	Yes
Bandwidth (>5 kHz)	Yes	Yes	Yes	No (100 Hz)	Good (10 kHz)	No (~1 kHz)
Small targets (1 × probe ∅)	Yes	No	Ok (spot size 10 µm)	No	Yes	Yes
Distance from target needed	No	No	Yes	No	No	No
Dependence on material type (metal, dielectric, etc.)	No	Yes	No	No	No	No
Dependence on surface reflectivity	No	Yes	Yes	No	No	No
Immunity to EMI ^2^	Yes	Yes	Yes	No	Yes	Yes
Operation in dirty/dusty environments	No	Good	Poor	Yes	Yes	Excellent
Measurement of exterior features not visible to non-contact sensors	No	No	No	Yes	Yes	Yes

^1^ Linear variable transformer; ^2^ electromagnetic interference.

## Data Availability

Data are not publicly available due to privacy considerations.

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
