# Peer review of "A Micron-Range Displacement Sensor Based on Thermo-Optically Tuned Whispering Gallery Modes in a Microcapillary Resonator"

_sensors, 2022, doi:10.3390/s22218312_

Round 1
Reviewer 1 Report
The present study proposes a micron- range displacement sensor based on a WGM microcapillary resonator filled with a nematic LC and a magnetic nanoparticles coated fiber half-taper where the LC temperature variation leads to a change in the refractive index. In general, the study is well presented and the experimental work was planned and aimed to demonstrate the features of the sensor
However, the authors listed three applications where the proposed sensor can be useful for, however, the reviewer thinks that this is not enough to show for the reader that proposed sensor has practical application. It was of prime importance to pick one specific application and highlight what is the value that proposed sensor can add for the picked application and to prove this experimentally for the reader. Can the authors comment on this?
Secondly, in the literature review section, the authors have to compare in more details their WGM displacement sensor to the other manuscripts that WGM resonator-based displacement sensors were the topic of their study. This will help the reader understands what is the gap that the authors are bridging by their work beside that the proposed sensor is the first WGM to utilize the photothermal effect
Reviewer 2 Report
A micron-range displacement sensor based on thermo-optically tuned whispering gallery modes in a microcapillary resonator
Authors: Zhe Wang et al
This manuscript deals with a scheme previously implemented thermos-optic tuning of a microcapillary resonator filled with a nematic liquid crystal and a fiber half-taper coated with a thin layer of magnetic nanoparticles placed inside the resonator. The authors’ claims that the proposed scheme shows that the above structure can be utilized as a displacement sensor with high sensitivity of 14 pm/mm. They also shown that the proposed sensor offers good repeatability, 73 small hysteresis and a competitive response time of 260 ms.
This paper is timely and contributes significant information to the ongoing study of WGM resonator-based displacement sensors utilizing pulling or bending. While the concept of a novel tunable WGM capillary resonator filled with a nematic LC has been dealt with in other publications, the authors provide a significant step forward by exploiting the robustness mechanical stability and ease of manipulation and adjustmen.
As the minor suggestions, authors may also address in the introduction the demands for experimental study and its implication of your results for recent
development in quantum technology.
I think it might be great to think about recent works on recent theoretical proposals on superior performances of novel types of quantum probes in precision measurement like in the https://www.mdpi.com/1099-4300/23/10/1353.
Overall the paper is well written and a nice contribution to the literature on the photo thermal effect within a WGM microcapillary to realize a displacement sensor and should be interesting for the community who are studying these.
I am happy to recommend the paper to be published in Sensors.
Reviewer 3 Report
Its a really nice piece or scientific research, and very original.
Author Response
We are very grateful for this comment and appreciation of our work.
Reviewer 4 Report
The article is well written. It regards a displacement sensor based on fiber optic technology by means of thermo-optical effect. Within the manuscript, experimental data are compared with numerical simulation.
In my opinion, a couple of comments may further improve the contents:
- From line 53, before to report some state-of-art WGM-based displacement sensors, it might be easier for the reader to have more information about the WGM technique.
- Starting with section 2, it is a bit hard for the reader to follow the sensor fabbrication process. For an easier understanding, a table or a flowchart may be the solution.
- Please give more information about the devices employed (e.g. BBS power and FWHM)
- I don't understand the test made in function of the pump laser power: did you make this test to enhance that the sensor is working also for lower power value? or the aim was the fit shown in figure 5b (in my opinion 4 points are too few, by the way)?
- Moreover, if during the sensor lecture the power is changing, how can you ensure that it is working correctly if the resonance wavelength is changing as well?
- At line 186 you wrote about repeatibility test reported in figure 4a: just 2 tests at different power level are not enough to test the repeatibility of the sensor. You should have done cyclic tests, invastigating on the variance, std and residuals.
Round 2
Reviewer 1 Report
The revised manuscript is now ready for publishing and the reviewer has no further comments